# KGQA-STAR: DELIBERATIVE REASONING FOR PLANNING ON KNOWLEDGE GRAPHS FOR QUESTION ANSWERING

## ABSTRACT

Large language models (LLMs) excel in natural language processing but struggle with knowledge-intensive tasks such as multi-hop reasoning, where issues like outdated knowledge, hallucinations, and weak planning often arise. Knowledge Graph Question Answering (KGQA) offers a promising solution but existing approaches face limitations in reasoning efficiency, graph structure utilization, and symbolic query generation. We propose KGQA-Star, a reinforcement learning(RL) enhancement framework that enhances LLM reasoning over knowledge graphs. KGQA-Star introduces a simplified symbol retrieval statement-KG retrieval plan and a symbol retrieval system (KGSRS) supporting error feedback and reflective correction. To address the lack of RL methods in KGQA, we build a high-quality KG-Cot dataset via data distillation and apply curriculum learning for cold-start training. The framework employs a three-stage process—exploration, planning, and reflection—optimized with Reinforce++ and stage-specific rewards. Experiments show that KGQA-Star significantly improves symbolic query quality and reasoning accuracy in complex KGQA tasks, offering a practical path to strengthen LLM performance in knowledge-intensive scenarios.

## 1 INTRODUCTION

With the rapid advancement of Large Language Models (LLMs), they have demonstrated remarkable capabilities in natural language processing (NLP) (Besta et al., 2024; Brown et al., 2020; Chowdhery et al., 2023). Through pretraining on billions or even hundreds of billions of parameters and learning from massive, diverse corpora (Rawte et al., 2023; Touvron et al., 2023), LLMs achieved strong performance in zero-shot and few-shot scenarios (Touvron et al., 2023; Min et al., 2022). However, LLMs still face significant challenges in knowledge-intensive tasks (Khot et al., 2022; Petroni et al., 2021). Their internal knowledge is static and prone to obsolescence (Sun et al., 2024; Wen et al., 2024), which leads to knowledge gaps and erroneous reasoning on up-to-date or domain-specific informatio (Fu et al., 2024; Chuang et al., 2024). Furthermore, the high cost of training and continuously updating LLMs makes knowledge synchronization even more difficult (Kandpal & Raffel, 2025; Jiang et al., 2025).

To address these issues, incorporating external knowledge bases has become a promising direction. Knowledge Graphs (KGs) provide structured, reliable, and interpretable factual information to support reasoning (Luo et al., 2025; LUO et al., 2024; Pan et al., 2024; Ma et al., 2024), making Knowledge Graph Question Answering (KGQA) a representative task in this context (Zhang et al., 2024; Jiang et al., 2023b). Nevertheless, existing KGQA approaches still suffer from limitations: step_by_step reasoning methods struggle with error accumulation and global reasoning efficiency (Guo et al., 2023; Ye et al., 2022; Sun et al., 2024), while retrieval-then-answer methods (Saxena et al., 2020; Wang et al., 2022; Park et al., 2023), especially symbolic retrieval (Luo et al., 2024b; LUO et al., 2024), often rely solely on supervised fine-tuning (SFT) and lack structured hints from the KG for specific queries, which makes it difficult to generate high-quality queries and further hinders reflection and error correction once retrieval fails. These challenges highlight the need for more effective reasoning and planning mechanisms in KGQA. At the same time, reinforcement learning (RL) has shown potential in aligning LLMs with task-specific objectives (OpenAI,

2024; Yang et al., 2025; Zhao et al., 2024b), improving planning, and enabling reflective correction in complex reasoning tasks, yet its application to KGQA remains underexplored.

In this work, we propose **KGQA-Star**, a reinforcement learning enhancement framework aimed at enhancing LLM inference on knowledge graphs to generate high-quality symbol retrieval statements. KGQA-Star introduces a simplified symbolic query representation, the KG Retrieval Plan ($KGRP$), along with a KG symbol retrieval system ($KGSRS$) that provides explicit error feedback. To address the lack of RL methods in KGQA, we build a high-quality KG-Cot dataset through data distillation and apply curriculum learning for cold-start training. Our framework further adopts a three-stage reinforcement learning process—exploration, planning, and reflection—optimized via Reinforce++ and stage-specific reward functions. Experiments on multiple KGQA benchmarks (Yih et al., 2016; Talmor & Berant, 2018; Petrochuk & Zettlemoyer, 2018) demonstrate that KGQA-Star significantly improves symbolic query quality and reasoning accuracy, offering a practical path to strengthen LLM performance in knowledge-intensive scenarios.

## 2 RELATED WORK

**Knowledge Graph Question Answering (KGQA).** KGQA aims to answer natural language questions by reasoning over structured knowledge graphs (KGs) (Zhang et al., 2024; Jiang et al., 2023b). Current research mainly follows two paradigms. The *step-by-step paradigm* (Jiang et al., 2023a; Kim et al., 2023) leverages local KG structures and provides interpretable reasoning but suffers from error accumulation in multi-hop reasoning and difficulty in capturing global connections (Guo et al., 2023; Ye et al., 2022; Sun et al., 2024). The *retrieval-then-answer paradigm* (Saxena et al., 2020; Wang et al., 2022; Park et al., 2023) first retrieves subgraphs or triples, then uses them as prompts for LLM reasoning. Neural retrieval encodes triples and questions into vectors for matching (Park et al., 2023), while symbolic retrieval fine-tunes LLMs to produce executable queries such as pattern graphs (Chen et al., 2021) or SPARQL statements (Chen et al., 2024b; Luo et al., 2024a;b; LUO et al., 2024).

**Reinforcement Learning for LLM Reasoning.** Reinforcement Learning (RL) has emerged as an effective approach to enhance LLM reasoning (OpenAI, 2024; Yang et al., 2025; Zhao et al., 2024b), improving decision-making through task-oriented rewards beyond SFT (Gunjal et al., 2025; Liu et al., 2025b). RL has shown success in domains such as mathematics, programming, and logical reasoning (Xie et al., 2025; Chen et al., 2024a; Liu et al., 2025a), as well as retrieval from unstructured text (Song et al., 2025; Chen et al., 2025). However, its integration with structured KGQA remains limited, particularly in methods that use RL to generate and optimize symbolic retrieval plans and enable structured reflection and error correction after execution.

## 3 METHODOLOGY

Compared with previous approaches (Tan et al., 2025a; Ao et al., 2025b), our model focuses on generating high-quality retrieval plans(symbol retrieval statements). The overall framework of KGQA-Star is shown in Figure 1. In this section, we first introduce the preliminaries of the KGQA domain, followed by detailed descriptions of the four major components of the framework.

### 3.1 PRELIMINARY

A knowledge graph (KG) stores factual knowledge as a set of triples $G = (h, r, t) \mid h, t \in E, \ r \in R$, where $E$ is the entity set and $R$ the relation set. By capturing the topological structure of entities and relations, KGs reveal complex semantic networks and provide explicit logical constraints for reasoning tasks (Ao et al., 2025a). A path in $G$ is $e_0 \xrightarrow{r_1} e_1 \xrightarrow{r_2} \dots \xrightarrow{r_l} e_l$, with $e_i \in E$ and $r_i \in R$ (Ji et al., 2024). The $n$-hop neighborhood of entity $e$ is $\mathcal{N}^n(e) = v \in E \mid \text{dist}(e, v) \leq n$, and the induced $n$-hop subgraph is $G_e^n = (h, r, t) \in G \mid h, t \in \mathcal{N}^n(e)$.

Knowledge Graph Question Answering (KGQA) seeks to map natural language questions to structured knowledge for accurate answering (Tan et al., 2025b; Toroghi et al., 2025). Formally, given a question $Q$ and a KG $G$, the objective is to learn $f : \mathcal{Q} \times G \rightarrow A$, where $A = e_a, e_a \in E$ is the answer set. We assume topic entities in $Q$ have been identified through entity linking, yielding the known set $E_Q$.

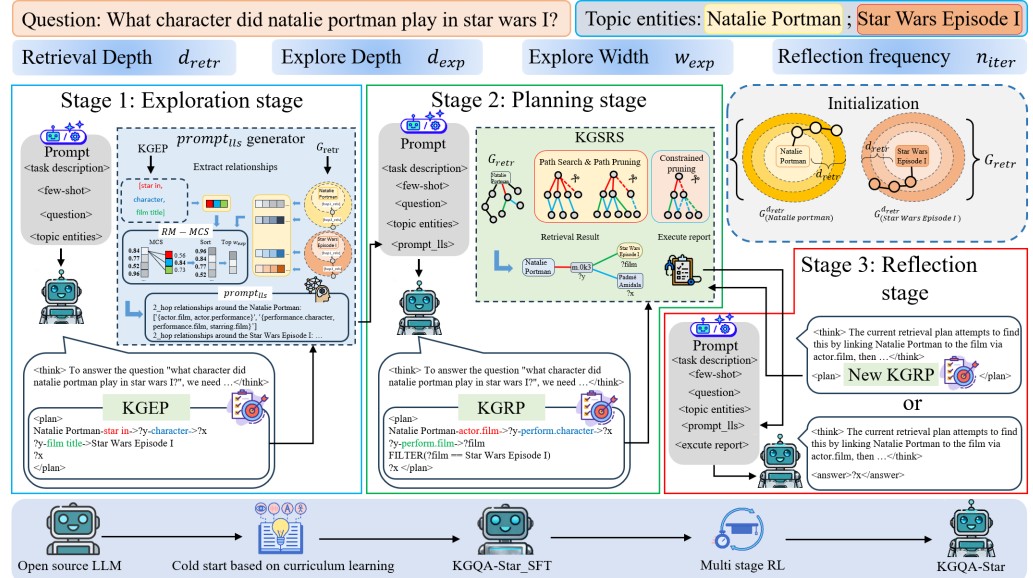

Figure 1: KGQA-Star framework. Given a question and topic entities, KGQA-Star proceeds in three stages: (1) Exploration—LLM generates a KGEP, from which The $promt_{lls}$ generator extracts local logical structure prompts from it through RM-MCS; (2) Planning—LLM produces a retrieval plan (KGRP), executed by KGSRS to obtain results and reports; (3) Reflection—LLM revises the plan or outputs the final answer based on execution feedback. In addition, KGQA-Star has undergone a cold start based on curriculum learning and three-stage reinforcement learning.

## 3.2 KG RETRIEVAL PLAN & KGRS

Mainstream KG symbol retrieval statements such as SPARQL and Cypher require strict syntactic correctness, which LLMs often fail to meet due to their stochastic nature, causing execution failures even for minor errors (Xia et al., 2025; Chen et al., 2023). Previous approaches employed supervised fine-tuning (SFT) to teach LLMs to generate simplified symbolic retrieval statements, improving the model's ability to produce valid queries, but these systems could only return subgraphs or entities and did not provide explicit error feedback. To overcome these limitations, we propose a new symbolic retrieval representation—the KG Retrieval Plan (KGRP)—along with its dedicated retrieval system, KGSRS.

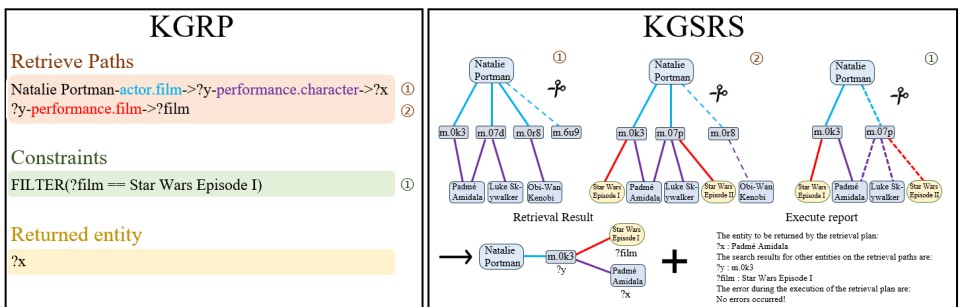

Figure 2: KGRP Example and KGSRS Operation Process.

### 3.2.1 KG RETRIEVAL PLAN

The KGRP originates from SPARQL but is more concise, readable, and generalizable to diverse KGs. It also offers strong extensibility, allowing easy modification by LLMs during the reflection stage. As shown in Figure 2, a KGRP consists of three parts: retrieval paths, optional constraints,

and the target entity to return. In this paper, we use $KGRP^*$ to represent the ground true retrieval plan for a questions.

**Retrieval Paths.** At least one retrieval path must be specified. Each path describes a relation chain from a starting entity to another entity, in the form:

$$head\ entity - relation1 -> intermediate\ entity - relation2 -> ... -> tail\ entity \quad (1)$$

Entities may be either topic entities $e_q$ from question $Q$ or unknown entities represented by placeholders such as $?x, ?y$. Subsequent paths allowed to reference previously introduced entities. Multiple paths are separated by newlines. For instance, the retrieval path in Figure 2 specifies that "Natalie Portman" reaches an intermediate node ?y via the relation actor.film; ?y is then linked to Natalie Portman's role name via performance.character, and simultaneously linked to ?film (the movie entity) via performance.film. Finally, the plan requires ?film to be "Star Wars Episode I."

**Constraints.** Constraints are optional and used to filter or rank retrieval results. Each constraint occupies a separate line.

**Return Entities.** The retrieval plan must explicitly indicate the target entities to return, specified by their identifiers.

### 3.2.2 KGSRS

The KG Symbolic Retrieval System (KGSRS) executes KGRPs over KG $G$ by sequentially processing statements, recording errors, and returning both retrieval results and error information. This enables LLMs to refine reasoning during the reflection stage and decide whether to revise the plan or provide an answer.

**Path Retrieval and Pruning.** KGSRS first clusters the interconnected retrieval paths to obtain $\{sub\_paths\}$ (Algorithm 5), and then organizes the retrieval results of each $sub\_paths$ into a retrieval plan tree, where nodes correspond to entities and edges to relations. Each path execution proceeds hop by hop from topic entities, with relation matching relaxed via semantic similarity when exact matches are absent. Invalid branches—e.g., those shorter than required hop length or failing to reach target entities—are pruned through DFS. If a required entity in the path is not found, KGSRS records the error and terminates the retrieval for that path. After each path retrieval is completed, it will return a set of triplets and pruned $head\ entity$ to be saved in the retrieval plan tree for further pruning.

**Constraint Pruning.** For each constraint statement, KGSRS filters entities in the retrieval plan tree and prunes branches that do not satisfy the condition.

The final execution report $report_{exe}$ summarizes retrieved entities and intermediate results, together with structured error messages (Table 9). This dual output not only improves retrieval robustness under imperfect queries but also provides explicit signals for reflection-driven plan revision (Figure 2).

### 3.3 THREE-STAGE REASONING FRAMEWORK

The three-stage reasoning framework constitutes the core of KGQA-Star (as illustrated in Figure 1). Unlike conventional approaches (Luo et al., 2024b; He et al., 2025), KGQA-Star does not directly rely on the LLM to generate the final retrieval query. Instead, it decomposes the reasoning process into three steps: first, during the exploration stage, the LLM generates an exploration plan to acquire structural information surrounding the topic entities $E_Q$, which serves as a local logical structure prompt $prompt_{lls}$; next, in the planning stage, the LLM reasons based on $prompt_{lls}$ and generates the $KGRP$. Then $KGSRS$ executes the plan and returns an execution report. Finally, in the reflection stage, LLM decides to either self correct or output the final answer..

Prior to reasoning, KGQA-Star performs initialization. For each topic entity $e_a$, it constructs an $n_hop$ subgraph centered on $e_a$, forming the retrieval subgraph $G_{retr}$, where $n = d_{retr}$ is a hyperparameter representing the retrieval depth of KGQA-Star. All subsequent retrieval plans are executed over $G_{retr}$.

$$G_{retr} = \{G_{e_q}^{(d_{retr})} \mid e_q \in E_Q\} \quad (2)$$

### 3.3.1 STAGE 1: EXPLORATION STAGE

High-quality KG structural prompts enable the LLM to better capture the local logical structure of the KG during reasoning, thereby substantially improving the quality of the generated retrieval queries. However, most existing KGQA methods either neglect this step or adopt inefficient per-hop prompting as in step_by_step reasoning paradigms. To address this issue, we explicitly design an exploration stage prior to query generation.

**Exploration plan generation.** The LLM, given the instruction, few-shot examples, question $Q$, and the topic entity set $E_Q$, produces a reasoning process and a KG exploration plan $KGEP$. Here, the few-shot examples are composed of the most similar prior questions and their associated $KGRP^*$, providing guidance on KG logical structures.

$$KGEP = LLM(instruction_{stage1}, few-shot, Q, E_Q) \tag{3}$$

**Local logical structure prompt extraction.** Although $KGEP$ follows the same format as the KG retrieval plan in section3.2.1, it is not directly executed since it is generated with limited quality based on few-shot guidance. Instead, we design a relation-matching algorithm based on maximum cross similarity, denoted as $RM-MCS$, to extract the local logical structure prompt $prompt_{lls}$ from $KGEP$. The definition of cross similarity $MCS$ is provided in Algorithm 1. Specifically, the $promptlls$ generator first extracts all relations from $KGEP$, denoted as $rels_{refer} = extract_r els(KGEP)$. For each topic entity $e_q$, $promptlls$ generator extracts $\mathcal{R}^n(e_q) \mid n = 1, \ldots, n_{sim}$ from $G_{retr}$. For each $\mathcal{R}^n(e_q)$, we compute its maximum cross similarity with $rels_{refer}$ as $rel_{sim} = MCS(\mathcal{R}^n(e_q), rels_{refer})$, and retain the top-$k$ relations with the highest $MCS$-$\mathcal{R}_k^n(e_q)$. Consequently, $promptlls$ consists of a relation subset surrounding each topic entity $e_q$, where $n_{sim} = d_{exp}$ represents the exploration depth and $k = w_{exp}$ represents the exploration width, both being hyperparameters.

$$prompt_{lls} = \{\{\mathcal{R}_{w_{exp}}^n(e_q) \mid n = 1, \ldots, d_{exp}\} \mid e_q \in E_Q\} \tag{4}$$

---

**Algorithm 1** Maximum Cross Similarity

---

**Require:** Embedding model $M$; candidate list $\mathcal{C}$; reference list $\mathcal{R}$
**Ensure:** List of maximum similarity scores **s**
1: **if** $\mathcal{C} = \emptyset$ **then**
2:    **return** $\emptyset$
3: **end if**
4: $\mathbf{E}_c \leftarrow M.\text{encode}(\mathcal{C})$                                                    ▷ candidate embeddings
5: $\mathbf{E}_r \leftarrow M.\text{encode}(\mathcal{R})$                                                    ▷ reference embeddings
6: $\mathbf{S} \leftarrow \text{cosine\_similarity}(\mathbf{E}_c, \mathbf{E}_r)$
7: **for** each candidate $c_i \in \mathcal{C}$ **do**
8:    $s_i \leftarrow \max_j \mathbf{S}[i, j]$
9: **end for**
10: **return** $\{s_1, s_2, \ldots, s_{|\mathcal{C}|}\}$

---

### 3.3.2 STAGE 2: PLANNING STAGE

In the planning stage, the LLM receives $prompt_{lls}$ along with $Q$, $E_Q$, and few-shot examples, and outputs the $KGRP$:

$$KGRP = LLM(instruction_{stage2}, few-shot, Q, E_Q, prompt_{lls}) \tag{5}$$

With the aid of $prompt_{lls}$, the LLM's outputs become more reliable. Moreover, the LLM can reason from a global perspective, which significantly improves reasoning efficiency compared to stepwise paradigms. Subsequently, $KGSRS$ executes $KGRP$ and generates an execution report $report_{exe}$.

### 3.3.3 STAGE 3: REFLECTION STAGE

The LLM refines its reasoning based on $report_{exe}$, either accepting retrieved entities as the answer to $Q$ or regenerating $KGRP$. Each new $KGRP$ leads to a new $report_{exe}$, which is then reintroduced into the LLM for further reflection. Reflection may iterate up to $n_{iter}$ times (default = 1). This mechanism balances answer reliability with computational cost.

$$KGRP \text{ or } E = LLM(instruction_{stage3}, few-shot, Q, E_Q, prompt_{lls}, report_{exe}) \quad (6)$$

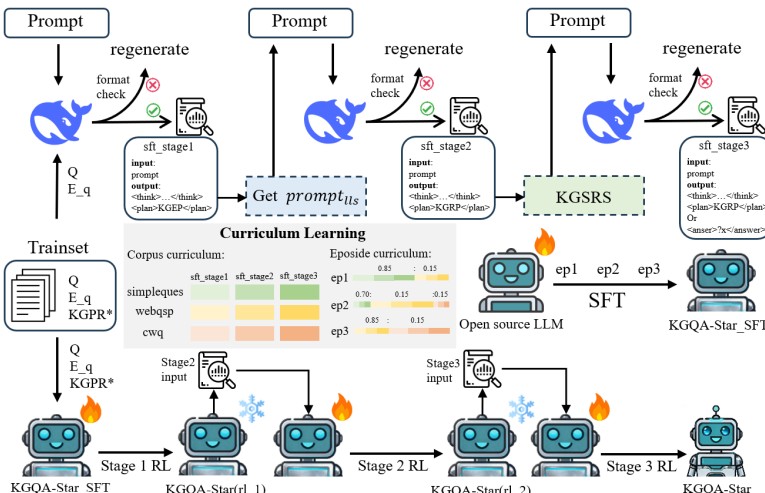

Figure 3: Cold start based on curriculum learning and three-stage reinforcement learning process. We distill stage-wise data with DeepSeek-R1, design curriculum-based cold-start fine-tuning (3 eps) to obtain KGQA-Star_SFT, and then perform three-stage RL training with tailored reward functions, freezing the model after each stage to generate data for the next.

### 3.4 HIGH QUALITY COT DATA CONSTRUCTION

Since no high-quality CoT dataset exists for KGQA, we construct KGQA-CoT via data distillation with DeepSeek-R1 (DeepSeek-AI, 2025) and subsequent rule-based filtering (Wang et al., 2025; Liu et al., 2025a). The dataset enables curriculum learning–based cold-start training for open-source LLMs (Gandhi et al., 2024).

**Data sources.** We adopt three benchmark KGQA datasets—SimpleQuestions (Petrochuk & Zettlemoyer, 2018), WebQSP (Talmor & Berant, 2018), and CWQ (Yih et al., 2016)—as the foundation of KGQA-CoT. The statistics of each dataset are shown in Table 4. A portion of their training set is extracted for data distillation, while the remaining data is retained for reinforcement learning.

**Data distillation and filtering.** Distillation focuses on teaching the LLM to produce reasoning traces and valid $KGRP$ in the required format $< think > \cdots < /think >< plan > \cdots < /plan >$ or $< think > \cdots < /think >< answer > \cdots < /answer >$. Outputs failing format or logical constraints are regenerated, ensuring high-quality structured data.

**Curriculum learning for cold start.**

Following evidence that curriculum learning surpasses random shuffling (Wang et al., 2021), we organize training data from simple to complex queries. This progression enables efficient cold-start learning, with allocation per epoch illustrated in Figure 3.

### 3.5 REINFORCEMENT TRAINING METHOD

In this study, we adopt **REINFORCE++** (Hu et al., 2025) as the reinforcement learning algorithm. Building on the classical REINFORCE framework, REINFORCE++ introduces baseline correction, variance reduction, and dynamic normalization strategies, effectively mitigating training instability

while preserving algorithmic simplicity (Wan et al., 2025). Compared to PPO, REINFORCE++ is less sensitive to hyperparameters and more robust in sparse and delayed reward scenarios.

We design rule-based reward functions tailored to different reasoning stages.

**Stage 1.** First, we extract all relations $rel_{pre}$ from $KGEP$ using the function $extract_rels()$. We then extract relations $rel_{refer}$ from $KGRP^*$. Following the approach in section 3.3.2, we compute maximum cross similarity (MCS), take the average, and obtain $rel\_sim_{av} = mean(MCS(rel_{pre}, rel_{refer}))$. In addition, we introduce a penalty term $p$ for mismatched relation counts:

$$p = \frac{|\,|rel_{pre}| - |rel_{true}|\,|}{\max\left(|rel_{pre}|,\ |rel_{true}|\right)} \tag{7}$$

The reward function for Stage 1 is defined as follows:

$$R_{\text{stage1}} = \begin{cases} (1 - p) * rel\_sim_{av}, & \text{correct format} \\ 0, & \text{incorrect format} \end{cases} \tag{8}$$

**Stage 2 and Stage 3.** We evaluate the predicted entity set $E_{pre}$ against the answer entity set $A$ using the F1 score:

$$\text{recall} = \frac{|E_{pre} \cap A|}{|A|} \tag{9}$$

$$\text{precision} = \frac{|E_{pre} \cap A|}{|E_{pre}|} \tag{10}$$

$$f_1 = \frac{2 \cdot \text{precision} \cdot \text{recall}}{\text{precision} + \text{recall}} = \frac{2 \cdot |E_{pre} \cap A|}{|E_{pre}| + |A|} \tag{11}$$

$Epre$ is the set of entities returned by executing $KGRP$. The reward functions for Stage 2 and Stage 3 are defined as follows:

$$R_{\text{stage2,3}} = \begin{cases} 0, & \text{incorrect format} \\ 0.1, & f_1 < 0.1 \\ f_1, & \text{other} \end{cases} \tag{12}$$

Each stage of reinforcement learning training reinforces the corresponding stage of the reasoning framework. After completing the training of each stage, we freeze the LLM parameters and let it generate the inputs required for the next stage of training.

## 4 EXPERIMENT

We aim to address three research questions: RQ1. How does KGQA-Star perform on standard KGQA benchmarks? RQ2. Does reinforcement learning and three-stage reasoning framework contribute to the overall performance of KGQA-Star? RQ3. Can KGQA-Star generate high-quality interpretable retrieval plans?

Our evaluation is conducted on three Freebase-based datasets: SimpleQuestions (Petrochuk & Zettlemoyer, 2018) (single-hop), WebQSP (Talmor & Berant, 2018), and CWQ (Yih et al., 2016) (multi-hop). We compare against five groups of baselines: non-LLM methods(no llm), closed-source LLMs(only llm), and LLM–KG approaches with either step_by_step reasoning(SR) or retrieval–answer paradigms(RA). Following prior work, we report exact-match accuracy (Hits@1) (Tan et al., 2025b; Sun et al., 2023) as the main metric. We always use Qwen2.5-7b-Instruct as the backbone LLM. Additional experimental settings, including dataset statistics, baseline configurations, and implementation details, are provided in the Appendix.

### 4.1 HOW DOES KGQA-STAR PERFORM ON KGQA TASKS?

From Table 1, KGQA-Star shows substantial gains on both single- and multi-hop tasks, improving Hits@1 by 5.1% on WebQSP and 7.3% on SimpleQuestions. On the challenging CWQ dataset,

Table 1: Performance comparison of different methods on Multi-Hop KGQA (CWQ, WebQSP) and Single-Hop KGQA benchmarks.

| Method | Class | LLM | Multi-Hop KGQA | | Single-Hop KGQA |
| --- | --- | --- | --- | --- | --- |
| | | | CWQ | WebQSP | SimpleQuestions |
| *Without external knowledge* | | | | | |
| KV-Mem | no llm | - | 18.4 | 46.7 | - |
| GraftNet | no llm | - | 36.8 | 66.4 | - |
| SR+NSM | no llm | - | 50.2 | 69.5 | - |
| UniKGQA | no llm | - | 50.7 | 75.1 | - |
| HGNet | no llm | - | 65.3 | 71.7 | - |
| CoT | only llm | GPT-3.5-Turbo | 38.8 | 66.2 | 20.3 |
| SC | only llm | GPT-3.5-Turbo | 45.4 | 71.1 | 18.9 |
| *With external knowledge* | | | | | |
| ToG/ToG-R | SR | GPT-4 | 69.5 | 86.2 | 66.7 |
| EiffQA | SR | GPT-4 | 69.5 | 82.9 | 76.5 |
| KG-Cot | SR | GPT-4 | 62.3 | 84.9 | 86.1 |
| EtD | RA | ChatGPT | 62.0 | 82.5 | - |
| ROG | RA | Qwen2.5-7b | 65.1 | 84.5 | - |
| KGQA-Star | RA | Qwen2.5-7b | **72.2** | **91.3** | **93.4** |

although KGSRS restricts the search space by matching only the most similar relation at each hop, KGQA-Star still outperforms TOG by 1.7%, confirming its effectiveness. Moreover, methods equipped with external knowledge generally outperform those without, highlighting the importance of external knowledge for LLMs. At the same time, retrieval–answer paradigms tend to underperform stepwise reasoning methods on multi-hop datasets, whereas KGQA-Star, through its three-stage reasoning and reinforcement learning, enhances the quality of symbolic retrieval statements and thus obtains superior results from the KG.

## 4.2 DOES REINFORCEMENT LEARNING AND THREE-STAGE REASONING FRAMEWORK CONTRIBUTE TO THE OVERALL PERFORMANCE OF KGQA-STAR?

Next, we evaluate whether reinforcement learning and three-stage reasoning framework indeed improve KGQA-Star's planning and reasoning (RQ2). Concretely, we compare KGQA-Star_SFT (curriculum-based cold-start), intermediate RL checkpoints, and the final KGQA-Star on the same test splits; to limit inference cost we randomly sample 200 instances per dataset. To isolate the

Table 2: The ablation experiments for KGQA-Star's three-stage reasoning.

| Method | CWQ | WebQSP | SimpleQuestions |
| --- | --- | --- | --- |
| KGQA-Star w/ $KGEP$ | 71.0 | 90.0 | 95.0 |
| KGQA-Star w/ $Q$ | 65.0 | 80.0 | 85.0 |
| KGQA-Star w/o stage1 | 64.0 | 53.5 | 74.7 |
| KGQA-Star w/o stage3 | 66.6 | 89.0 | 95.0 |
| KGQA-Star | 71.5 | 91.7 | 96.0 |

contributions of the three-stage framework, we also run targeted ablations. For reflection stage, we compare: KGQA-Star w/o stage3 (no reflection stage). For exploration, we compare: KGQA-Star w/ $Q$ (where $prompt_{lls}$ is obtained via question-based $MCS$), KGQA-Star w/ $KGEP$ (where $prompt_{lls}$ is obtained via whole $KGEP$-based $MCS$) and KGQA-Star w/o stage1 (exploration stage omitted, no $prompt_{lls}$).

As shown in Fig. 4, RL consistently improves KGQA-Star's planning and reasoning across all datasets compared with the supervised-only baseline (KGQA-Star_SFT). Performance increases steadily through intermediate RL stages, with the complete three-stage RL model achieving the best overall results. Complementing this, the stage-wise ablations in Table 2 show that removing the exploration stage results in a substantial performance drop, underscoring the importance of problem-specific local logic structural prompts over the KG. Furthermore, KGQA-Star w/$Q$ indicates that without leveraging the LLM's internal knowledge for exploration, high-quality $prompt_{lls}$ cannot be obtained, while KGQA-Star w/$KGEP$ demonstrates that our proposed $RM - MCS$ algorithm yields more fine-grained exploration outcomes. Finally, KGQA-Star w/o stage 3 illustrates

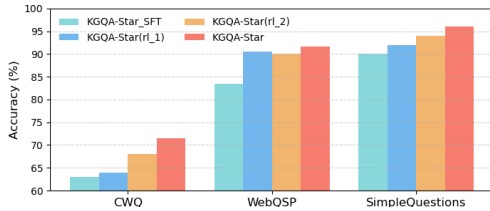 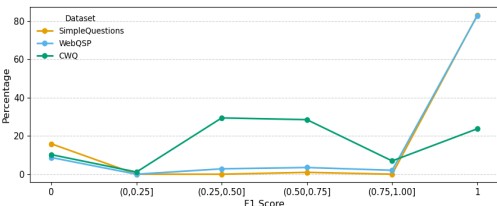

Figure 4: Performance comparison of KGQA-Star trained using RL and SFT on three QA benchmarks.

Figure 5: F1 score distribution of the relationships in $KGRP$ and $KGRP^*$ on three QA benchmarks.

that the LLM benefits from reflecting on and correcting errors through the $report_{exe}$ returned by $KGSRS$.

### 4.3 CAN KGQA-STAR GENERATE HIGH-QUALITY INTERPRETABLE RETRIEVAL PLANS?

To further evaluate the quality of the retrieval plans ($KGRP$) generated by KGQA-Star, we extract the relation sets $rel_{pre}$ and $rel_{refer}$, following the procedure in Section 3.5, and compute their F1 score according to Eq. equation 11. The F1 score distributions across the three datasets are shown in Fig. 5, where more than 80% of the retrieval plans on SimpleQuestions and WebQSP are identical to the gold standard $KGRP$, while for CWQ, the majority of the plans achieve an F1 score above 50%. In addition, Table 3 presents a case study: although the plan generated during the exploration stage is not fully identical to $KGRP$, KGQA-Star refines it through $prompt_{lls}$ and $report_{exe}$, ultimately producing a correct retrieval plan and making the right decision in the reflection stage. More case studies are provided in the Appendix.

Table 3: Example of high-quality three-stage reasoning for KGQA-Star.

| Question | what did mark zuckerberg study? |
|---|---|
| **Topic entity** | Mark Zuckerberg |
| **Answer entity** | Computer programming; Computer Science; Psychology |
| **Stage1 output** | $< think >$ To answer the question "what did Mark Zuckerberg study?", we need to find ... $< /think >< plan >$ Mark Zuckerberg-person.education-→?y-education.institution-→?x-topic.notable_types-→College/University ?y-education.field_of_study-→?z ?z $< /plan >$ |
| **Stage2 output** | $< think >$ The question is asking what Mark Zuckerberg studied, which is related to ... $< /think >< plan >$ Mark Zuckerberg-person.education-→?y-education.major_field_of_study-→?x ?x $< /plan >$ |
| **Stage3 output** | $< think >$ The retrieval plan correctly uses these relationships to find the tail entity ?x, which represents the major field(s) of study. The execution result shows that ?x is "Computer Science ; Psychology ..." and there are no errors in the retrieval process. Therefore, the current retrieval plan is sufficient to answer the question accurately. $< /think >< answer >$?x$< /answer >$ |

## 5 CONCLUSION

We proposed **KGQA-Star**, a reinforcement learning–based framework that enhances LLM reasoning over knowledge graphs through a three-stage process of exploration, planning, and reflection. By introducing the KG Retrieval Plan (KGRP) and its symbol retrieval system (KGSRS), our approach enables explicit error feedback and interpretable symbolic reasoning. Experiments on multiple KGQA benchmarks show that KGQA-Star substantially outperforms existing baselines, improving both accuracy and query interpretability. Ablation studies confirm the contributions of each stage and the effectiveness of reinforcement learning in strengthening symbolic retrieval. Future work will explore broader knowledge sources and more advanced RL strategies to further extend the framework's applicability.

## ETHICS STATEMENT

This work proposes KGQA-Star, a framework designed to enhance the reasoning and planning capabilities of large language models (LLMs) in knowledge graph question answering (KGQA). We only use publicly available datasets (SimpleQuestions, WebQSP, and CWQ), each with proper licenses, and strictly follow privacy and compliance requirements. No personally identifiable sensitive information is involved in this research. Our primary goal is to promote interpretable and controllable symbolic reasoning methods, reducing hallucinations and misleading outputs from LLMs in knowledge-intensive tasks. We encourage researchers to carefully consider ethical and societal risks when applying related technologies to avoid potential misuse.

## REPRODUCIBILITY STATEMENT

To ensure reproducibility, we provide a detailed description of the KGQA-Star framework, including its architecture, training procedure, three-stage reasoning process, and reward function design. The datasets used in our experiments (SimpleQuestions, WebQSP, CWQ) are all publicly available, and hyperparameter settings, evaluation metrics, and ablation study designs are fully documented in the main text and appendix. Upon acceptance, we plan to release the core code and the constructed KGQA-CoT dataset, enabling other researchers to reproduce and further verify our experimental results.

## USE OF LARGE LANGUAGE MODELS STATEMENT

This research indeed employs open-source large language models (e.g., Qwen2.5-7B) for reasoning, retrieval plan generation, and reflection optimization. All models are obtained from publicly accessible sources, and no closed-source or proprietary LLMs are used. During training and experimentation, we combined data distillation with reinforcement learning, and all results are derived solely from the methods described in this paper, without reliance on outputs from external, unspecified LLMs. This ensures the interpretability and controllability of our research conclusions.

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

# A APPENDIX

## A.1 PROMPT

We summarize the specific prompts used in the three-stage reasoning process, as shown in Figure 6.

**Stage1 Prompt**

You are a knowledge graph question-answering assistant. Now you need to reason based on the given question and topic entity, and produce a retrieval plan for querying the knowledge graph.
The retrieval plan must consist of three parts:
1. One or more retrieval paths (at least one is required). Each path is structured as:entity-relation->entity-relation->...->tail entity(unknown entities should be represented as '?' plus letter, such as "?x").If there are multiple paths, separate them by the newline character '\n'.
2. Optional filtering or sorting constraints, with line breaks between multiple constraints. For example:- FILTER() (e.g., FILTER (?from < ?end); FILTER (?num < "1983-01-03"))- ORDER BY (e.g., ORDER BY ?num LIMIT 1; ORDER BY DESC(?num) LIMIT 1 OFFSET 1)
3. The entity to be returned (this is mandatory), indicating which entity the retrieval plan is designed to return.
Separate these three parts by newline characters '\n'.

<Few shot>

Given the question: {}
and the topic entity/entities: {}
please reason out the retrieval plan.

Output your reasoning process wrapped with <think></think> tags, and the retrieval plan wrapped with <plan></plan> tags, in the format:<think>...</think><plan>...</plan>.Do not include anything other than <think>...</think><plan>...</plan>.

**Stage2 Prompt**

You are a knowledge graph question-answering assistant. Now you need to reason based on the given question , topic entity and the n-hop relationships around the topic entity, and produce a retrieval plan for querying the knowledge graph.
The retrieval plan must consist of three parts:
1. One or more retrieval paths (at least one is required). Each path is structured as:entity-relation->entity-relation->...->tail entity(unknown entities should be represented as '?' plus letter, such as "?x").If there are multiple paths, separate them by the newline character '\n'.
2. Optional filtering or sorting constraints, with line breaks between multiple constraints. For example:- FILTER() (e.g., FILTER (?from < ?end); FILTER (?num < "1983-01-03"))- ORDER BY (e.g., ORDER BY ?num LIMIT 1; ORDER BY DESC(?num) LIMIT 1 OFFSET 1)
3. The entity to be returned (this is mandatory), indicating which entity the retrieval plan is designed to return.Separate these three parts by newline characters '\n'.
The part of n-hop relationships around the topic entity are represented by '[{{one hop relationships}}, {{two hop relationships}}, {{three hop relationships}}, ...]', for reference during inference.

<Few shot>
Given the question: {}
and the topic entity/entities: {}
<$prompt_{lls}$>
please reason out the retrieval plan.

Output your reasoning process wrapped with <think></think> tags, and the retrieval plan wrapped with <plan></plan> tags, in the format:<think>...</think><plan>...</plan>.Do not include anything other than <think>...</think><plan>...</plan>.

**Stage3 Prompt**

You are a knowledge graph question-answering assistant. Now you need to reason based on the given question, topic entity/entities, and the n-hop relationships around the topic entity/entities, together with the existing retrieval plan and its execution result, to either output the final answer or propose a revised retrieval plan.The retrieval plan must consist of three parts:
1. One or more retrieval paths (at least one is required). Each path is structured as:entity-relation->intermediate entity-relation->...->tail entity(unknown entities should be represented as '?' plus letter, such as "?x").If there are multiple paths, separate them by the newline character '\n'.
2. Optional filtering or sorting constraints, with line breaks between multiple constraints. For example:- FILTER() (e.g., FILTER (?from < ?end); FILTER (?num < "1983-01-03"))- ORDER BY (e.g., ORDER BY ?num LIMIT 1; ORDER BY DESC(?num) LIMIT 1 OFFSET 1)
3. The entity to be returned (this is mandatory), indicating which entity the retrieval plan is designed to return.Separate these three parts by newline characters '\n'.
The part of n-hop relationships around the topic entity are represented by '[{{one hop relationships}}, {{two hop relationships}}, {{three hop relationships}}, ...]', for reference during inference.

<Few shot>
Given the question: {}
and the topic entity/entities: {}
<$prompt_{lls}$>
Retrieval Plan: {}
Execution Result: {}
please reason out.

Output your reasoning process wrapped with <think></think> tags, and if you propose a new retrieval plan, wrap it with <plan></plan> so the overall format is <think>...</think><plan>...</plan>, or if you return the final answer, wrap the returned entity with <answer></answer>.

Figure 6: Detailed prompt words used in the three-stage reasoning framework.

## A.2 ALGORITHM

The detailed algorithms of $KGSRS$ is presented as follows:

---

**Algorithm 3** Subgraph Multi-Path Retrieval

---

**Require:** Graph $G$, path set $\{P_1, P_2, \ldots, P_n\}$, model $M$
**Ensure:** Retrieval tree $R$, error list $\{E_1, \ldots, E_n\}$
 1: Initialize retrieval tree $R$
 2: **for** path $P_i$ **do**
 3:     Determine start entities (head entity or previous results)
 4:     Run **Single-Path Retrieval** on $P_i$
 5:     Insert retrieved triples into $R$
 6:     Update retrieval template tree
 7:     **if** retrieved entities mismatch **then**
 8:         Prune $R$
 9:     **end if**
10: **end for**
11: **return** $R, \{E_i\}$

---

---

**Algorithm 2** Single-Path Retrieval

---

**Require:** Knowledge graph $G$, path $P = [e_0, r_1, e_1, \ldots, r_k, e_k]$, start entities $S$, similarity model $M$
**Ensure:** Retrieved triples $T$, updated start entities $S'$, error list $E$
 1: Initialize $T \leftarrow \varnothing$, intermediate map $D$, error list $E$
 2: **for** $i = 0$ **to** $k - 1$ **do**
 3:     Set current nodes from $S$ or $D[e_i]$
 4:     **for all** node $u$ in current nodes **do**
 5:         Find successors $v$ with relation $r_{i+1}$
 6:         **if** no successors found **then**
 7:             Approximate relation using top-$k$ similarity from $M$
 8:         **end if**
 9:     **end for**
10:     **if** $e_{i+1}$ is variable **then**
11:         Update $D[e_{i+1}]$ with successors
12:     **else**
13:         Check constant match (including numeric match)
14:     **end if**
15:     Append matching triples $(u, r_{i+1}, v)$ to $T$
16:     **if** no successor found **then**
17:         Record error in $E$ and **break**
18:     **end if**
19: **end for**
20: Build retrieval tree from $T$ and filter reachable nodes
21: **return** $T, S', E$

---

**Algorithm 4** Whole-Plan Execution

---

**Require:** Subgraph data, query plan $Q$, mapping mid2name, model $M$
**Ensure:** execute report $report_{exe}$
 1: Parse $Q$ into path set, filter set, order set
 2: Call **Clustering Connected Retrieval Paths Build** $\{sub\_paths\}$
 3: **for** sub_paths **do**
 4:     Call **Subgraph Multi-Path Retrieval**
 5:     Collect entity mappings
 6: **end for**
 7: Apply filters: prune inconsistent branches
 8: Apply ordering: prune or reorder entities
 9: Aggregate final results
10: Generate execute report(result set $R$, error dict $E$)
11: **return** $report_{exe}$

---

To ensure end-to-end interpretability and scalability, we organize the retrieval pipeline into four tightly coupled sub-algorithms executed in a "cluster → execute → constrain" order.

First, Whole-Plan Execution(Algorithm 4) orchestrates the process: it parses the input plan into a set of retrieval paths, a set of constraints, and a designated return target, then invokes Clustering Connected Retrieval Paths(Algorithm 5) to group paths by structural connectivity. The clustering stage decomposes each path into triples and uses connectivity tests—handling both variable-headed and fixed-head paths, as well as identical begin-keys—to partition paths into disjoint connected subsets, thereby isolating cross-group dependencies and reducing execution complexity.

Whole-Plan Execution then iterates over clusters and calls Subgraph Multi-Path Retrieval(Algorithm 3) per cluster: within a cluster, paths are executed and reconciled around shared entity assignments, with all matched triples written incrementally into a cluster-level retrieval tree; any inconsistency triggers pruning of violating branches.

Each individual path is expanded by Single-Path Retrieval(Algorithm 2), which performs stepwise edge matching ⟨u, r, v⟩ from the current frontier; when exact relations are missing, a similarity model is used for relation approximation, and constant tails support numeric equality checks. After

---

**Algorithm 5** Clustering Connected Retrieval Paths

---

**Require:** Path list $P = \{p_1, p_2, \ldots, p_n\}$, optional flag $res$
**Ensure:** Clustered subgraphs paths $C$
1: Initialize $ans\_dict \leftarrow \emptyset$, $num \leftarrow 0$
2: **for** each path $p_i \in P$ **do**
3:     Convert $p_i$ into a list of triples
4:     **if** $num = 0$ **then**
5:         Create new subgraph with $p_i$
6:         $num \leftarrow num + 1$
7:     **else**
8:         **if** head entity of $p_i$ is variable **then**
9:             **for** each subgraph $sg \in ans\_dict$ **do**
10:                 **if** $p_i$ fully connected with $sg$ **then**
11:                     Merge $p_i$ into $sg$, update indices
12:                     **break**
13:                 **end if**
14:             **end for**
15:         **else**
16:             $cur\_begin \leftarrow$ first entity-relation of $p_i$
17:             **if** exists $sg \in ans\_dict$ with same $begin$ **then**
18:                 Create new subgraph with $p_i$
19:                 $num \leftarrow num + 1$, **continue**
20:             **end if**
21:             $merged \leftarrow$ False
22:             **for** each subgraph $sg \in ans\_dict$ **do**
23:                 **if** $p_i$ fully connected with $sg$ **then**
24:                     Merge $p_i$ into $sg$, update indices
25:                     $merged \leftarrow$ True, **break**
26:                 **end if**
27:             **end for**
28:             **if** not $merged$ **then**
29:                 Create new subgraph with $p_i$
30:                 $num \leftarrow num + 1$
31:             **end if**
32:         **end if**
33:     **end if**
34: **end for**
35: Initialize $ans \leftarrow \emptyset$, $ans\_num \leftarrow 0$
36: **for** each cluster $sg \in ans\_dict$ **do**
37:     Add corresponding original paths into $ans$
38:     Normalize paths (if $res = True$)
39:     $ans\_num \leftarrow ans\_num + |sg.indexes|$
40: **end for**
41: **if** $ans\_num \neq |P|$ **then**
42:     **raise error** "wrong!"
43: **end if**
44: **return** $ans$

---

all clusters are processed, Whole-Plan Execution applies plan-level constraints: it selects, re-ranks (including LIMIT/OFFSET), and back-propagates pruning on the retrieval trees to maintain structural consistency.

Finally, it returns the target entities and a retrieval report summarizing cluster assignments, per-cluster/per-path diagnostics( 9), and the effects of constraint application. This layered design propagates local correctness to global consistency while enabling efficient execution.

### A.3 TRAINING PROCESS

We provide the process information of each training stage.

Curriculum-based Cold Start. The loss curve of the LLM in this stage is shown in Figure 7.

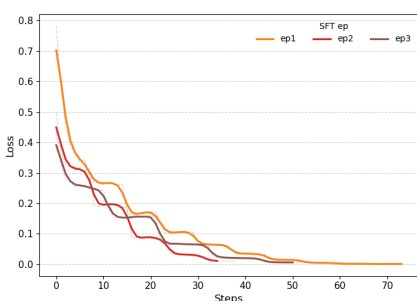

Figure 7: Loss curve during cold start phase.

It can be observed that each curriculum-trained LLM converges, with initial training loss values consistently decreasing, which verifies the rationality of the curriculum design.

Multi-stage Reinforcement Learning. We show the changes in rewards(Figure 8) and model response lengths(Figure 9) across different training stages. Figure show that the average reward steadily increases across all three reinforcement learning stages: stage1 achieves the highest mean reward (0.894), followed by stage2 (0.878) and stage3 (0.875), with relatively low variance, indicating stable convergence. This progression suggests that RL optimization effectively enhances the quality of symbolic queries and reasoning strategies, particularly in the earlier stages where reward gains are most pronounced. In parallel, response length exhibits a clear downward trend as training progresses. The mean response length decreases from 243 tokens in stage1 to 235 in stage2, and further to 200 in stage3. This reflects that KGQA-Star gradually learns to generate more concise yet informative retrieval plans, avoiding unnecessary expansions. Together, these results demonstrate that three-stage RL training not only improves reasoning quality (as evidenced by rising rewards) but also enhances efficiency by reducing redundant reasoning steps. This dual effect highlights the effectiveness of our curriculum design, where exploration promotes structural awareness, planning consolidates symbolic reasoning, and reflection refines outputs for precision and brevity.

### A.4 EXPERIMENT

#### A.4.1 SPECIFIC EXPERIMENTAL SETUP

Experimental Datasets. To evaluate KGQA-Star on KGQA tasks, we conducted experiments on three KBQA datasets: two multi-hop datasets (CWQ, WebQSP) and one single-hop dataset (SimpleQuestions). For large-scale datasets CWQ and SimpleQuestions, following prior work (Tan et al., 2025b; Sun et al., 2023), we randomly sampled 1,000 test cases from each. All datasets are based on Freebase as the background knowledge graph, which contains approximately 88 million entities, 20,000 relations, and 126 million triples. Detailed dataset statistics are provided in Table 4.

Baselines. We compare RoG with twelve baselines, grouped into four categories: (1) methods without LLMs, (2) pure LLM reasoning methods, (3) LLM+KGs with stepwise reasoning paradigms, and (4) LLM+KGs with retrieval–answer paradigms. Baseline details are as follows:

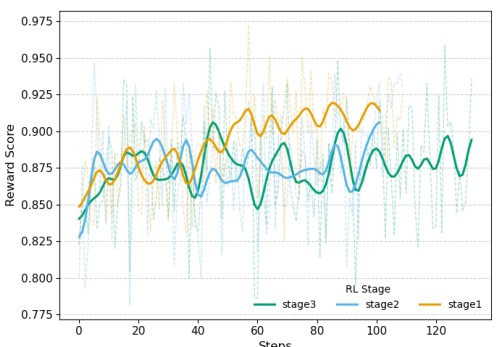
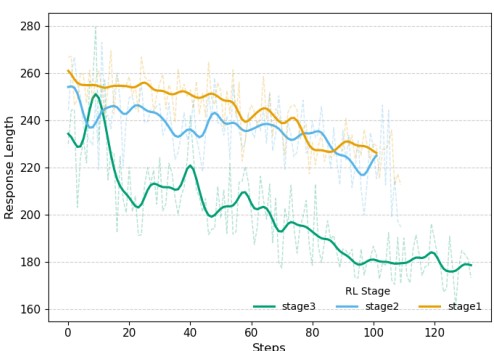

Figure 8: The reward score curve of three-stage reinforcement learning.

Figure 9: The response length curve of three-stage reinforcement learning.

Table 4: Statistics of the datasets.

| Dataset | Answer Format | Train | Test |
|---|---|---|---|
| SimpleQuestions | Entity/Number | 14,894 | 21664 |
| WebQSP | Entity/Number | 3098 | 1639 |
| CWQ | Entity/Number | 27734 | 3531 |

- **KV-Mem** (Miller et al., 2016): Utilizes key-value memory networks to store triples and performs multi-hop reasoning by iteratively operating on memory.

- **GraftNet** (Sun et al., 2018): Retrieves relevant subgraphs from the KG via entity linking.

- **SR+NSM** (Zhang et al., 2022): Proposes relation-path–based subgraph retrieval for multi-hop reasoning.

- **UniKGQA** (Jiang et al., 2023b): Unifies graph retrieval and reasoning into a single model integrated with PLMs, achieving state-of-the-art KGQA performance.

- **HGNet** (Chen et al., 2023): An end-to-end model that performs hierarchical autoregressive decoding to generate query graphs.

- **GPT-3.5-Turbo+CoT**: Uses chain-of-thought prompting to enhance reasoning.

- **GPT-3.5-Turbo+SC**: Uses self-consistency (SC) prompting.

- **TOG** (Sun et al., 2024): Treats the LLM as an agent that interactively explores related entities and relations in the KG and reasons over retrieved knowledge.

- **EiffQA** (Dong et al., 2025): Leverages LLM commonsense ability for global planning, introducing an enhanced query strategy tightly coupled with KG constraint-based pruning.

- **KG-Cot** (Zhao et al., 2024a): Employs small-scale stepwise graph reasoning models for reasoning over KGs and generates high-confidence knowledge chains for large-scale LLMs.

- **EtD** (Liu et al., 2024): Integrates LLMs with graph neural networks (GNNs) for KG reasoning.

- **RoG** (Luo et al., 2024): Fine-tunes LLMs to generate KG-based relation paths as faithful plans, which are then used to retrieve valid reasoning paths for faithful LLM reasoning.

**Experimental Settings.** We adopt Qwen2.5-7b-Instruct as the backbone LLM. First, we perform curriculum-based cold start on the distilled dataset KGQA-Cot, with three epochs in total, each consisting of five training cycles. The batch size is set to 2, and the learning rate to 1e-5. For reinforcement learning, we use the REINFORCE++ algorithm in three stages, with one epoch per stage, a batch size of 64, and 8 rollouts. Training is conducted on eight H20-96G GPUs. During inference, the retrieval depth is set to 4, and exploration depth varies by dataset: 1 for SimpleQuestions, 2 for WebQSP, and 4 for CWQ. The exploration breadth is set to 3, and the number of reflection iterations defaults to 1. We use bge-large-en as the embedding model, with a 1024-dimensional embedding size.

### A.4.2 Further Analysis

In this section, we provide a further analysis of the experimental results obtained in 4.1. We mainly report KGQA-Star's performance on questions involving multiple topic entities, multi-hop reasoning, and constrained conditions. Additionally, we report the average inference time of Qwen2.5-7b within KGQA-Star.

**F1 score.** We further evaluate KGQA-Star using the F1 score, computed as in Eq. equation 11. Results are presented in Table 5.

Table 5: Comparison of F1 score of KGQA-Star on three QA benchmarks..

| Dataset | F1 Score |
|---|---|
| SimpleQuestions | 93.3 |
| WebQSP | 90.1 |
| CWQ | 64.6 |

As shown in Table 5, KGQA-Star achieves high F1 scores on SimpleQuestions and WebQSP, further confirming the quality of the generated $KGRP$s. On the more challenging CWQ dataset, performance drops to 64.6%, primarily because the retrieval space is severely constrained and the 7B model occasionally hallucinates when handling complex multi-hop queries. Nevertheless, KGQA-Star still surpasses our baseline RoG (56.2%), demonstrating its effectiveness even under difficult reasoning scenarios.

**Multiple Topic Entities.** Table 6 shows performance stratified by the number of topic entities. For single-entity questions ($|E_Q|$=1), KGQA-Star achieves strong results on SimpleQuestions (93.3%) and WebQSP (91.1%), with lower accuracy on CWQ (65.7%). Adding a second entity improves performance on WebQSP (92.1%) and CWQ (81.6%), reflecting the benefit of entity constraints in narrowing the search space. With three entities, WebQSP reaches 100%, while CWQ drops slightly to 77.1%, indicating that excessive compositional complexity can offset these benefits.

Table 6: Performance comparison of KGQA-Star with different numbers of topic entity problems on three QA benchmarks.

| Dataset | $|E_Q|$ | | |
|---|---|---|---|
| | 1 | 2 | 3 |
| SimpleQuestions | 93.3 | - | - |
| WebQSP | 91.1 | 92.1 | 100 |
| CWQ | 65.7 | 81.6 | 77.1 |

**Multi-hop Reasoning.** Table 7 shows that KGQA-Star's performance fluctuates as the number of hops increases. On SimpleQuestions (primarily single-hop), hits@1 is 93.3% when $|rel_{true}| = 1$, consistent with its single-hop nature. In WebQSP, single-hop performance is highest (94.9%), but hits@1 declines for longer hops (e.g., 83.4% for two-hop, 88.8% for four-hop, and 33.3% for six-hop), reflecting the difficulty of complex reasoning chains. By contrast, CWQ, as a multi-hop dataset, shows greater robustness: 77.5% for two-hop, 65.8% for four-hop, 46.6% for six-hop, with performance rebounding at seven- and eight-hop questions (75.0% and 92.5%, respectively). This indicates that KGQA-Star has some long-chain reasoning capacity but faces bottlenecks at intermediate chain lengths, likely due to search space expansion and path uncertainty.

Table 7: Performance comparison of KGQA-Star with different numbers of relations problems on three QA benchmarks.

| Dataset | $|rel_{true}|$ | | | | | | | |
|---|---|---|---|---|---|---|---|---|
| | 1 | 2 | 3 | 4 | 5 | 6 | 7 | 8 |
| SimpleQuestions | 93.3 | - | - | - | - | - | - | - |
| WebQSP | 94.9 | 83.4 | 92.5 | 88.8 | - | 33.3 | 0.0 | - |
| CWQ | 80.0 | 77.5 | 73.3 | 65.8 | 67.0 | 46.6 | 75.0 | 92.5 |

**Constrained Statements.** Table 8 further analyzes performance on constrained questions. Hits@1 reaches 65.1% on WebQSP and 74.5% on CWQ. Compared to unconstrained situations, constrained

problems are more challenging because they require semantic understanding of conditions and precise KG filtering, while KGQA-Star requires generating accurate constraint statements. KGQA-Star performs relatively better on CWQ, potentially benefiting from more training examples with constraints. Overall, these results highlight remaining challenges in handling complex logical constraints.

Table 8: Performance comparison of KGQA-Star with constrained statement problems on three QA benchmarks.

| Dataset | constrained statement problems |
|---|---|
| SimpleQuestions | - |
| WebQSP | 65.1 |
| CWQ | 74.5 |

**Inference and Execution Time.** Figure 10 reports the average inference time across datasets and the average execution time of retrieval plans. Overall, SimpleQuestions requires the shortest average time, WebQSP is slightly longer, and CWQ is the longest. This trend aligns closely with dataset complexity: SimpleQuestions mainly contains single-hop problems with short reasoning paths, WebQSP includes a proportion of multi-hop queries, and CWQ involves complex multi-hop and constrained problems, requiring longer reasoning.

Table 9: The defined error message and its corresponding type.

| Error Type | Error description |
|---|---|
| Retrieve Path Error | Head node of the retrieval path {} is unknown! |
| | Head node of the retrieval path {} is not in KG! |
| | Entity {} was not retrieved. |
| | Exceeding the hop count range. Unable to retrieve entity {}. |
| Constraint Statement Error | Parse fail! Format error in constraint statements. |
| | Not found {}! |
| | All variables {} do not meet the condition! |
| | No limit number! |
| | Variable {} cannot be sorted! |

**Error Analysis.** We conducted a statistical analysis of the errors in the execution of the retrieval plans generated by KGQA Star on three datasets. We separately counted the number and proportion of the two types of errors in Table 9. The result is shown in Figure 11. In CWQ, retrieval path errors are relatively higher than in other datasets, while constraint-related errors dominate (66.3%), underscoring the difficulty of combining multi-hop inference with logical filtering. WebQSP exhibits very few path errors (0.6%) but still suffers from a high proportion of constraint errors (69.8%), indicating that KGQA-Star handles short and medium reasoning paths effectively but still struggles with constraint generation. For SimpleQuestions, constraint errors appear extremely high (98.7%) because the dataset lacks constraint-based questions; thus, any generated constraints lead to execution failures, though without affecting final retrieval outcomes. After further observation, we found that most constraint statement errors are due to KGQA-Star generation format errors, which cannot execute. This inspires our next optimization steps.

### A.4.3 ADDITIONAL ABLATION STUDY

We conduct ablation studies for deeper insights into experimental results.

**Impact of the Reflection Stage.** We evaluate the contribution of reflection stage by comparing four variants: KGQA-Star without stage 3 (KGQA-Star w/o stage3), KGQA-Star with a single reflection ($n_{iter}$=1), and KGQA-Star with multiple reflections ($n_{iter}$=3, 5). Results are reported in Table 10. Moderate reflection improves performance: introducing one reflection yields substantial gains over no reflection (e.g., +4.9 on CWQ), and three reflections provide slight further improvements. However, excessive reflections ($n_{iter}$=5) degrade performance, particularly on the multi-hop CWQ dataset, where hallucinations become more pronounced. These results suggest that reflection is beneficial when used sparingly, but over-reflection may amplify reasoning noise instead of correcting errors.

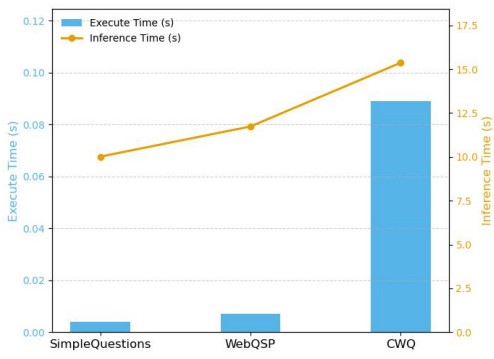

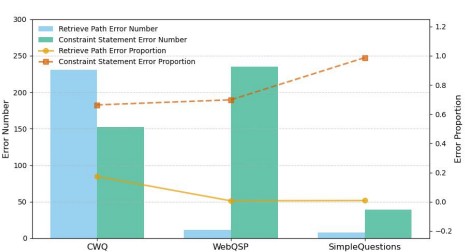

Figure 10: Inference and execution Time.          Figure 11: Erro Anaylsis.

Table 10: Performance comparison of KGQA-Star using different reflection frequency on three QA benchmarks.

| Method | CWQ | WebQSP | SimpleQuestions |
|---|---|---|---|
| KGQA-Star w/o stage3 | 66.6 | 89.0 | 95.0 |
| KGQA-Star $n_{iter} = 1$ | 71.5 | 91.7 | 96.0 |
| KGQA-Star $n_{iter} = 3$ | 72.0 | 92.0 | 96.0 |
| KGQA-Star $n_{iter} = 5$ | 66.0 | 93.0 | 96.0 |

**Impact of Exploration Breadth.** We further investigate the effect of exploration breadth $w_{exp}$ by evaluating KGQA-Star under three settings: $w_{exp} \in 1, 3, 5$. Results are reported in Table 11. As $w_{exp}$ increases, each hop of the local logical structure prompt $prompt_{lls}$ incorporates more candidate relations, enabling the model to consider a broader range of reasoning paths. This significantly enhances reasoning ability, with the most pronounced improvements observed on the CWQ dataset, where Hits@1 increases from 62.4 at $w_{exp} = 1$ to 73.0 at $w_{exp} = 5$. Performance on WebQSP remains relatively stable, while SimpleQuestions benefits moderately, indicating that broader exploration is especially valuable for complex multi-hop reasoning.

Table 11: Performance comparison of KGQA Star with different exploration breadth on three QA benchmarks.

| Method | CWQ | WebQSP | SimpleQuestions |
|---|---|---|---|
| KGQA-Star $w_{exp} = 1$ | 62.4 | 86.0 | 95.0 |
| KGQA-Star $w_{exp} = 3$ | 71.5 | 91.7 | 96.0 |
| KGQA-Star $w_{exp} = 5$ | 73.0 | 91.0 | 98.0 |

**Test-Time Scaling.** We evaluate KGQA-Star with Self-Consistency (SC), where the model samples $N$ answers per question and returns the majority entity. As shown in Table 12, larger $N$ improves performance on the challenging CWQ dataset ($71.5 \rightarrow 74.0$ at $N=15$), while gains on WebQSP are marginal ($91.7 \rightarrow 92.0$) and SimpleQuestions saturates around $N=10$ ($96.0 \rightarrow 97.0$). These results suggest that SC effectively reduces stochastic errors in complex multi-hop reasoning but provides limited benefit for simpler tasks. Improvements diminish beyond $N=5$–$10$, where additional cost outweighs accuracy gains, indicating this as a practical trade-off point.

### A.4.4 CASE STUDY

We also present two illustrative examples in the figures: one shows KGQA-Star solving a multi-hop multi-entity question, and the other demonstrates reflective correction of a reasoning plan.

This query involves both multi-hop reasoning and multi-entity constraints, making it a challenging test case. In the exploration stage, given the topic entities President of the United States and World War II, the system performed cross-similarity matching to identify relevant relations and generated local subgraph prompts ($prompt_{lls}$) as candidate reasoning paths. Building on these results, the planning stage constructed a structured query plan that linked presidents to their terms in office,

Table 12: Performance comparison of KGQA-Star using SC strategy on three QA benchmarks.

| Method | CWQ | WebQSP | SimpleQuestions |
|---|---|---|---|
| KGQA-Star (N=1) | 71.5 | 91.7 | 96.0 |
| KGQA-Star (N=5) | 72.0 | 91.6 | 97.0 |
| KGQA-Star (N=10) | 72.4 | 92.0 | 97.0 |
| KGQA-Star (N=15) | 74.0 | 92.0 | 96.5 |

constrained these terms by the start and end dates of World War II, and applied ordering to select the earliest-serving president whose term overlapped with the war period. The query executed successfully, yielding Franklin D. Roosevelt as the answer, with retrieved temporal information confirming that his presidency (1933–1945) overlapped with World War II (1939–1945). In the reflection stage, the model re-examined the execution results, verified that the identified entity satisfied the temporal and semantic constraints of the query, and confirmed Franklin D. Roosevelt as the final answer. This case study demonstrates how the exploration stage expands the search space with relevant relations, the planning stage produces precise and interpretable reasoning paths with appropriate constraints, and the reflection stage ensures correctness, collectively enabling KGQA-Star to handle complex multi-hop, multi-entity questions with accuracy and interpretability.

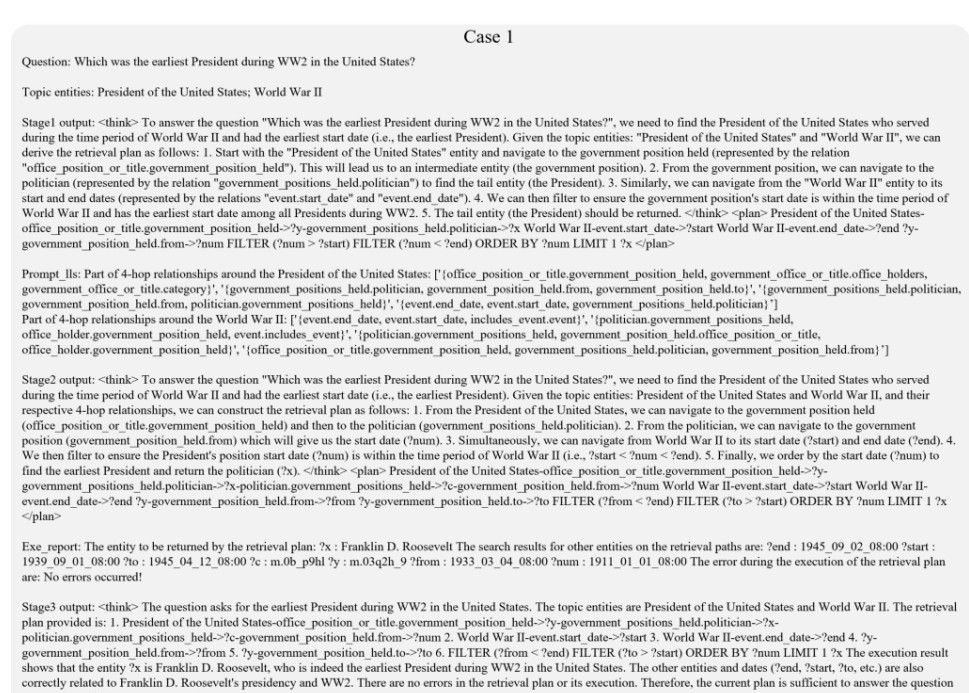

Figure 12: KGQA-Star answers multi hop multi entity questions.

To illustrate the impact of the reflection stage in KGQA-Star, we present a case involving the question: "What team's mascot is named Hank?", which requires correcting an initially flawed retrieval plan. The topic entity is Hank, and in the exploration stage, the model identified relevant 4-hop relationships including team_mascot.sports_team and sports_team.team_mascot, generating candidate reasoning paths ($prompt_{lls}$). During the planning stage, a preliminary retrieval plan was constructed connecting Hank to a candidate sports team ?c and further to ?x representing the mascot. Execution of this plan returned ?c as "Brewers Roster", which reflected a specific team roster rather than the intended sports team, highlighting a limitation in the initial plan. In the reflection stage, KGQA-Star re-evaluated the execution output, detected the discrepancy, and modified the retrieval plan to explicitly ensure that the correct sports team associated with the Hank mascot was returned. This involved applying additional constraints to distinguish between specific rosters and the general team entity, ultimately producing the correct answer. This example demonstrates how the reflection stage allows KGQA-Star to detect and correct errors in the retrieval plan, improving reasoning robustness

---

Case 2

Question: What team's mascot is name Hank?

Topic entities: Hank

Stage1 output: <think> The question asks for the team whose mascot is named Hank. The topic entity is Hank. To find the answer, we need to find a path from the topic entity Hank to the unknown entity (the team) using the relation team_mascot.sports_team. This will lead us to the tail entity (the team) which is the entity to be returned. </think> <plan> Hank-team_mascot.sports_team->?c ?c </plan>

Prompt_lls: Part of 4-hop relationships around the Hank: ['{team_mascot.sports_team, mascot.team, topic.notable_for}', '{sports_team.team_mascot, team.mascot, sports_team.sport}', '{team_mascot.sports_team, mascot.team, topic.notable_types}', '{sports_team.team_mascot, team.mascot, sports_team.sport}']

Stage2 output: <think> The question asks for the team whose mascot is named Hank. The topic entity is Hank. The given 4-hop relationships around Hank include 'team_mascot.sports_team' and 'sports_team.team_mascot', which directly relate to the question. To find the retrieval path: 1. Start with the topic entity Hank. 2. Using the 'team_mascot.sports_team' relationship, we can connect Hank to an unknown sports team entity ?c. 3. Then, using the 'sports_team.team_mascot' relationship, we can connect ?c to the tail entity ?x, which represents the team's mascot (Hank). 4. The entity to be returned is ?c, which is the sports team whose mascot is Hank. Since no additional filtering or sorting constraints are needed, the retrieval plan will simply return ?c. </think> <plan> Hank-team_mascot.sports_team->?c-sports_team.team_mascot->?x ?c </plan>

Exe_report: The entity to be returned by the retrieval plan: ?c : Brewers Roster The search results for other entities on the retrieval paths are: ?x : Bonnie Brewer ; Hank ... The error during the execution of the retrieval plan are: No errors occurred!

Stage3 output: <think> The question asks for the team whose mascot is named Hank. The topic entity is Hank, and the retrieval plan provided is: Hank-team_mascot.sports_team->?c-sports_team.team_mascot->?x ?c The execution result returns ?c as "Brewers Roster", which seems to be a specific team roster rather than the general sports team. This suggests that the current retrieval plan might be too specific or incorrect since it directly retrieves the roster instead of the sports team associated with the Hank mascot. To correct this, we should modify the retrieval plan to ensure we get the correct sports team rather than the roster. The relationships available include 'team_mascot.sports_team' and 'sports_team.team_mascot', which are correctly used, but we need to filter or specify constraints to ensure we get the correct team. A better retrieval plan would be to explicitly look for the sports team associated with the Hank mascot and ensure it is the correct team by filtering or constraints if necessary. </think> <plan> Hank-team_mascot.sports_team->?c-sports_team.team_mascot->?x FILTER (?x is the correct mascot) ?x </plan>

Figure 13: KGQA-Star corrects erroneous KGRP through a reflection phase.

and ensuring that multi-hop queries yield semantically accurate answers even when the initial plan is suboptimal.

---

