# OpenReview forum: "KGQA-Star:Deliberative Reasoning for Planning on Knowledge Graphs for Question Answering"
_ICLR.cc/2026/Conference — ICLR 2026 Conference Withdrawn Submission_

### Official Review · Reviewer_1EfA · 2025-10-28

**Soundness:** 2
**Presentation:** 2
**Contribution:** 2
**Rating:** 4
**Confidence:** 3

**Summary:**

This paper introduces KGQA-Star, a framework that aims to improve the ability of Large Language Models (LLMs) to reason over Knowledge Graphs (KGs) for question answering. The authors identify several key challenges in existing KGQA methods, including inefficient reasoning, poor use of graph structure, and difficulty in generating correct symbolic queries (like SPARQL). To address these issues, KGQA-Star proposes a novel three-stage reasoning process—Exploration, Planning, and Reflection—which is optimized using reinforcement learning (RL).

**Strengths:**

The application of reinforcement learning to the generation of symbolic KG queries is a novel and important contribution. Most prior work has relied solely on SFT, which can be brittle. Using RL with a system that provides explicit error feedback (KGSRS) allows the model to learn from its mistakes and improve its planning capabilities in a more robust way.

**Weaknesses:**

The KGQA-Star framework is quite complex, involving multiple stages of LLM generation, a custom retrieval system, a multi-stage training process with both SFT and RL, and several hyperparameters (e.g., retrieval depth, exploration depth/width). While the results are impressive, this complexity could be a barrier to adoption and reproduction. It would be helpful if the authors could comment on the overall system complexity and the engineering effort required to implement it.

The paper mentions creating a high-quality CoT dataset by distilling from DeepSeek-R1. This is a crucial step for the cold-start training. However, the details of this distillation process are a bit sparse. What were the prompts used to get DeepSeek-R1 to generate these traces? How extensive was the rule-based filtering? Since this dataset is a key prerequisite for the rest of the method, more detail on its construction and its general quality would be beneficial.

**Questions:**

see weaknesses

---

### Official Review · Reviewer_9nrJ · 2025-10-29

**Soundness:** 2
**Presentation:** 2
**Contribution:** 2
**Rating:** 4
**Confidence:** 4

**Summary:**

The paper proposed KGQA-Star, a reinforcement learning (RL) enhancement framework that enhances LLM reasoning over knowledge graphs. They introduced the KG Retrieval Plan (KGRP) and its symbol retrieval system (KGSRS), and proposed a three-stage process of exploration, planning, and reflection to enhance LLM’s reasoning over KGs.

**Strengths:**

- The proposed KG Retrieval Plan (KGRP)—along with its dedicated retrieval system, KGSRS—is more concise, readable, and generalizable to diverse KGs. It provides a good alternative to the existing SPARQL and Cypher.
- The experiments demonstrated that KGQA-Star achieved good performance on several KGQA problems.

**Weaknesses:**

- The system is very complicated. It requires interaction with the LLM through three stages: exploration, planning, and reflection. Additionally, it involves SFT and reinforcement learning at three different stages, as well as curriculum learning for a cold start. This makes reproduction difficult.
- The method is not novel. Except for the KGRP part, the three-stage pipeline—exploration, planning, and reflection—has been explored in related work such as RoG and ToG[1-2]. How does KGQA-Star differentiate itself from them?

\[1\] Think-on-Graph: Deep and Responsible Reasoning of Large Language Models on Knowledge Graphs. ICLR 2024.

\[2\] Plan-on-Graph: Self-Correcting Adaptive Planning of Large Language Models on Knowledge Graphs. NeurIPS 2024.

**Questions:**

None

---

### Official Review · Reviewer_pCLK · 2025-10-31

**Soundness:** 2
**Presentation:** 1
**Contribution:** 2
**Rating:** 2
**Confidence:** 5

**Summary:**

This paper proposes a reinforcement learning (RL) enhancement framework to enhance LLM reasoning over knowledge graphs, termed KGQA-Star. To be specific, the proposed method introduces a simplified symbolic query representation, the KG retrieval plan, along with a KG symbol retrieval system that can provide explicit error feedback. To conduct RL training, this paper builds a high-quality KG-Cot dataset through data distillation and apply curriculum learning for cold-start training. The proposed method is optimized by three-stage RL process (exploration, planning, and reflection). Extensive experiments demonstrate the effectiveness of the proposed method.

**Strengths:**

1.	This paper presents an RL method for KGQA and also provides a KG-Cot dataset.
2.	Extensive experiments show the effectiveness of the proposed method.

**Weaknesses:**

1.	The figures could be further refined to enhance readability. In particular, the font size in Figures 1-3 may be quite small.
2.	The paper may lack some baseline methods for comparison, such as GNN-RAG [1], SubgraphRAG [2], and PoG [3].
3.	There are some typos in the paper. For example, in line 212, it should probably be written as $n_{hop}$. It would be beneficial to polish the manuscripts to improve readability.

[1] Mavromatis, Costas, and George Karypis. "Gnn-rag: Graph neural retrieval for large language model reasoning." arXiv preprint arXiv:2405.20139 (2024).

[2] Li, Mufei, Siqi Miao, and Pan Li. "Simple is Effective: The Roles of Graphs and Large Language Models in Knowledge-Graph-Based Retrieval-Augmented Generation." The Thirteenth International Conference on Learning Representations.

[3] Chen, Liyi, et al. "Plan-on-graph: Self-correcting adaptive planning of large language model on knowledge graphs." Advances in Neural Information Processing Systems 37 (2024): 37665-37691.

**Questions:**

Please see in Section **Weaknesses**

---

### Official Review · Reviewer_ZtWf · 2025-11-01

**Soundness:** 2
**Presentation:** 2
**Contribution:** 2
**Rating:** 2
**Confidence:** 4

**Summary:**

The paper presents KGQA-Star, a framework combining reinforcement learning and symbolic reasoning for knowledge-graph question answering. It introduces a structured retrieval plan(KGRP), a reasoning system (KGSRS), and a three-stage RL optimization (exploration, planning, reflection). Experiments on SimpleQuestions, WebQSP, and CWQ show performance gains over prior KGQA models.

**Strengths:**

1. A easy to understand three-stage reasoning framework.
2. The framework integrates symbolic planning with LLM-based reflection and further provides a KG-CoT dataset
3. It presents a reinforcement learning approach within the “retrieve-then-answer” paradigm for KGQA, exploring how RL can enhance reasoning quality and retrieval–generation alignment.

**Weaknesses:**

1.	The RL component provides limited methodological novelty. The paper mainly reuses Reinforce++ with minor modifications and heuristic stage-specific rewards without theoretical analysis. All defined rewards are matching-based(Eq. 7–12), with no consideration of efficiency-related factors such as reasoning depth or exploration cost. Could you clarify what specific innovations, beyond the use of REINFORCE++, are introduced? For example, are there any convergence guarantees, empirical analyses or ablation study on the reward design to validate the effectiveness and robustness of the proposed component?
2.	The paper defines the exploration depth and width as hyperparameters, but provides no analysis of how different depth settings affect performance. If the reflection stage identifies that the current exploration depth is insufficient, is there any mechanism to adapt or update exploration depth dynamically during training or inference？
3.	The paper mentions that training data are organized from simple to complex queries to facilitate curriculum-based cold-start learning in Sec. 3.4. However, the manuscript does not clarify how query complexity is defined or measured. Is this distinction based on hop count, entity count, linguistic length, or reasoning depth? Moreover, no ablation studies are provided to demonstrate that curriculum learning itself contributes to better convergence or final performance.
4.	There are several recent works closely related to this paper that are not discussed or compared against. For example, Plan-on-Graph [1] adopts a similar three-stage reasoning framework with adaptive planning and reflection, while KnowGPT [2] employs reinforcement learning to optimize knowledge extraction and reasoning. These works should be incorporated into the related work section and considered as experimental baselines for a fair comparison.
5.	The reference list contains a duplicated entry (the first two references), and the main text includes numerous typos that affect readability, suggesting insufficient proofreading and a lack of attention to presentation quality expected at a top-tier venue.

[1] Plan-on-Graph: Self-Correcting Adaptive Planning of Large Language Models on Knowledge Graphs, NeurIPS 2024.
[2] KnowGPT: Knowledge Graph based PrompTing for Large Language Models, NeurIPS 2024.

**Questions:**

Refer to the weaknesses part.

---

### Note · Authors · 2025-12-04

I have read and agree with the venue's withdrawal policy on behalf of myself and my co-authors.